# Neu3 Sialidase Activates the RISK Cardioprotective Signaling Pathway during Ischemia and Reperfusion Injury (IRI)

**DOI:** 10.3390/ijms23116090

**Published:** 2022-05-29

**Authors:** Marco Piccoli, Simona Coviello, Maria Elena Canali, Paola Rota, Paolo La Rocca, Federica Cirillo, Ivana Lavota, Adriana Tarantino, Giuseppe Ciconte, Carlo Pappone, Andrea Ghiroldi, Luigi Anastasia

**Affiliations:** 1Laboratory of Stem Cells for Tissue Engineering, IRCCS Policlinico San Donato, Piazza Malan 2, San Donato Milanese, 20097 Milan, Italy; marco.piccoli@grupposandonato.it (M.P.); simona.coviello@grupposandonato.it (S.C.); mariaelena.canali@studenti.unimi.it (M.E.C.); federica.cirillo@grupposandonato.it (F.C.); ivana.lavota@grupposandonato.it (I.L.); tarantino.adriana@hsr.it (A.T.); 2Institute for Molecular and Translational Cardiology (IMTC), San Donato Milanese, 20097 Milan, Italy; paola.rota@unimi.it (P.R.); paolo.larocca@unimi.it (P.L.R.); giuseppe.ciconte@grupposandonato.it (G.C.); carlo.pappone@grupposandonato.it (C.P.); 3Department of Biomedical, Surgical and Dental Sciences, University of Milan, Via Mangiagalli 31, 20097 Milan, Italy; 4Department of Biomedical Sciences for Health, University of Milan, Via Mangiagalli 31, 20097 Milan, Italy; 5Faculty of Medicine and Surgery, University Vita-Salute San Raffaele, Via Olgettina 58, 20097 Milan, Italy; 6Arrhythmology Department, IRCCS Policlinico San Donato, Piazza Malan 2, San Donato Milanese, 20097 Milan, Italy

**Keywords:** sialidase-3, *Neu3*, ischemia and reperfusion injury, gangliosides, myocardial infarction, cardioprotective strategies

## Abstract

Coronary reperfusion strategies are life-saving approaches to restore blood flow to cardiac tissue after acute myocardial infarction (AMI). However, the sudden restoration of normal blood flow leads to ischemia and reperfusion injury (IRI), which results in cardiomyoblast death, irreversible tissue degeneration, and heart failure. The molecular mechanism of IRI is not fully understood, and there are no effective cardioprotective strategies to prevent it. In this study, we show that activation of sialidase-3, a glycohydrolytic enzyme that cleaves sialic acid residues from glycoconjugates, is cardioprotective by triggering RISK pro-survival signaling pathways. We found that overexpression of *Neu3* significantly increased cardiomyoblast resistance to IRI through activation of HIF-1α and Akt/Erk signaling pathways. This raises the possibility of using Sialidase-3 activation as a cardioprotective reperfusion strategy after myocardial infarction.

## 1. Introduction

Cardiovascular diseases (CVDs) persist as the leading cause of morbidity and mortality worldwide [1,2]. The most recent data from the National Institutes of Health show that 92.1 million adults in the United States (36.6% of the total population) have at least one type of cardiovascular disease and that 30.87% of deaths are attributable to this type of disease alone [3]. Among CVDs, acute myocardial infarction (AMI) is a life-threatening condition that requires rapid and successful intervention. Reperfusion strategies (with fibrinolytic or antithrombotic therapies) and primary percutaneous coronary intervention (P-IPC) represent the gold standard for AMI treatment. They can successfully rescue an ischemic myocardium eventually reducing mortality in infarcted patients [4,5]. However, rapid restoration of blood flow in an ischemic myocardium, although life-saving, inevitably leads to specific damage called ischemic and reperfusion injury (IRI), which triggers tissue degeneration that leads to heart failure [6]. It is known that numerous mechanical extracellular and intracellular processes are involved in the pathogenesis of IRI. For example, inflammation, neurohumoral activation, and oxidative stress have all been shown to be responsible for cardiac tissue damage [7,8]. Several approaches targeting these molecular pathways have been proposed to prevent cardiomyocyte death and promote cell survival. They include ischemic preconditioning (IPC) [9], ischemic postconditioning (IPost) [10,11], and remote ischemic conditioning (RIC) [12] as the most effective treatments to limit infarct size [13]. Due to the modest decreases in myocardial damage caused by these strategies, other approaches have been explored [14,15,16]. Their common goal is to trigger endogenous cardioprotective processes associated with specific molecular mechanisms known as the reperfusion injury salvage kinase (RISK) and survival activating factor enhancement (SAFE) pathways [17,18,19]. In particular, the RISK pathway is characterized by activation of the phospho-inositide 3-kinase (PI3K)-Akt and the mitogen-activated protein kinase (Mek)/extracellular signal-regulated kinase (Erk1/2) cascade [20] and has been shown to be fundamental against IRI. RISK regulates the opening of the mitochondrial permeability transition pore (mPTP), which is critical for reperfusion injury [21,22,23]. Inhibition of mPTP opening can be triggered by the heterodimeric transcription complex hypoxia inducible factor-1α (HIF-1α) [24], although this pathway had not been fully explored [25]. In view of our discovery that Sialidase-3 controls HIF-1α activation via a prolyl hydroxylase-independent signaling pathway [26], in this work we investigated the involvement of Sialidase-3 in the cardiac response to IRI.

## 2. Results

### 2.1. Sialidase-3 Modulation in a Mouse Model of IRI

Expression of *Neu3* was assessed in a mouse model of IRI obtained by 30-min ligation of the left anterior descending coronary artery followed by reperfusion (Figure 1A). Myocardial damage was validated by echocardiography and morphological analysis. Both ejection fraction (EF) and fractional shortening (FS) were significantly reduced after seven days of reperfusion as compared with sham animals (26.49% +/− 13.06 versus 58.47% +/− 2.796 and 12.44% +/− 6.518 versus 30.7% +/− 1.835, respectively) (Figure 1B,C). Masson’s trichrome staining of ventricular sections showed the formation of an extensive scar on the anterior ventricular wall characterized by a significant degree of fibrosis, confirming the induction of damage associated with ischemia and reperfusion (Figure 1D). The relative expression of *Neu3* was measured by Real Time PCR using mRNAs obtained from tissue samples of the infarct area, and compared with heart tissue from control mice that underwent cardiac surgery without ligation. Results showed that a 30-min LAD ligation caused a 40% decrease in *Neu3* expression in the infarct area compared with sham (Figure 1F). Reperfusion of the infarcted area resulted in progressive recovery of *Neu3* expression, reaching levels comparable to those of sham mice after four days (Figure 1F). No significant changes in *Neu3* expression were detected in the nonischemic tissue samples of the hearts of the same mice subjected to IRI (Figure 1G).

### 2.2. Effects on Cell Proliferation, Toxicity and Neu3 Expression of an In Vitro Model of IRI

To investigate the role of sialidase-3 during IRI, a simple in vitro model of IRI was developed in our laboratory by culturing H9c2 rat cardiomyoblasts at 1% O_2_ in DMEM without glucose for up to 12 h (*ischemic phase*), followed by a switch to complete growth medium under 21% O_2_ for up to 48 h (*reperfusion phase*) (Figure 2A). Control cells were cultured under normoxic conditions in a complete growth medium throughout the experiments. Results showed that the ischemic phase caused a significant cell loss and reduction in proliferation (−70% at 12 h, Figure 2B) along with a significant increase in cell cytotoxicity (+4-fold at 12 h, Figure 2C), with both effects reversed during reperfusion (Figure 2B,C).

Analysis of *Neu3* expression during the in vitro ischemic phase showed an initial upregulation (+69% and +23% at 1 and 3 h, respectively) followed by a progressive reduction that reached −70% at 12 h (Figure 3A). A parallel change of expression was observed for Sialidase-3 enzymatic activity, which was increased after 1, 3, and 6 h of ischemia (+1.3, +2.9, and +2.3-fold, respectively) before a marked decrease after 12 h (−63%) (Figure 3B). In the reperfusion phase, both sialidase *Neu3* expression and activity were restored to levels similar to those measured in control H9c2 cells (Figure 3A,B).

### 2.3. Effects of Sialidase Neu3 Overexpression in H9c2 Cardiomyoblasts

H9c2 cells were transduced with a lentiviral vector containing the coding sequence of rat sialidase *Neu3* to overexpress the enzyme stably and named as OX-Neu3 cells. Cardiomyoblasts were also infected with a scrambled lentiviral vector (the same lentiviral vector employed to up-regulate *Neu3* but not containing the coding sequence of rat Sialidase-3) and used as a control, hereafter named SCR. Among several clones tested (Appendix A), *Clone 3* was selected for further experiments since it had the highest sialidase *Neu3* expression, which was 15-fold higher than that of control cells, accompanied by an approximately threefold increase of sialidase-3 activity (Figure 4A,B). *Neu3* overexpression also caused a 70% decrease of ganglioside GM3, one of the main substrates of the sialidase-3 [26,27] (Figure 4C).

Control (SCR) and *Neu3*-overexpressing (OX-Neu3) H9c2 myoblasts showed superimposable proliferation activity (Appendix A). A 12-h ischemic phase caused significant cell loss (−62% as compared with time 0) in SCR cells, which was markedly reduced by *Neu3* overexpression (−25% as compared with time 0) (Figure 5A). OX-Neu3 myoblasts rapidly resumed an exponential proliferation profile during reperfusion, whereas SCR cells proliferated at a significantly lower rate and fell short of reaching their initial number after 60 h. Apoptosis analysis using the Caspase-Glo^®^ assay showed marked activation of caspases 3/7 in SCR cells after 3, 6, and 12 h of ischemia, which was three times higher than in OX-Neu3 cells. The extent of apoptosis decreased significantly in both groups during reperfusion, although caspase levels of SCR were still significantly higher (2-fold on average) than in OX-Neu3 cells, the latter being very similar to untreated controls (Figure 5B). DAPI staining of nuclei for detection of chromatin condensation and nuclear blebbing was significantly reduced in OX-Neu3 as compared to SCR cells (−70% on average) (Figure 5C,D).

To test whether the effects of *Neu3* overexpression could be reversed by inhibition of the enzyme, OX-Neu3 H9c2 were treated with a noncommercial sialidase inhibitor (LR332) that was recently reported to be a specific inhibitor of sialidase Neu3 [28]. Analysis of cell growth showed that treatment with LR332 did not alter the sensitivity of SCR cells to ischemia and reperfusion, as both LR332-treated and -untreated SCR showed a superimposable reduction in proliferation. On the contrary, treatment of OX-Neu3 cells with the inhibitor reversed the beneficial effects of sialidase-3 overexpression, as cell proliferation was significantly reduced upon exposure to IRI, similar to that observed in SCR controls. In addition, LR332 treatment of OX-Neu3 cells caused a fourfold increase in apoptosis compared with untreated cells, similar to SCR controls (Appendix A).

### 2.4. Sialidase Neu3 Up-Regulation Effects on the RISK Pathway and HIF-1α

To assess whether *Neu3* overexpression activates the RISK pathway, the ratio between the inactive and active (phosphorylated) forms of the pro-survival kinases Akt and Erk1/2 during IRI was measured by western blot (Figure 6A). The results showed activation of both kinases during the ischemia and reperfusion phases, which was on average twofold higher in OX-Neu3 cells than SCR controls (Figure 6B,C). Then, the stability of HIF-1α protein during the ischemic phase was analyzed by measuring the activation of the HIF-1α oxygen-responsive domain (ODD) with a specific luciferase assay. Overexpression of *Neu3* resulted in a twofold increase in HIF-1α stability after 1 h of ischemia that persisted throughout the ischemic phase (Appendix A). This resulted in a significant increase in protein levels in OX-Neu3 cells which was twofold higher than in SCR controls at all time points examined (Figure 6D). In these sets of experiments, SCR controls represent cells transduced with scramble lentiviruses and exposed to the same treatment as OX-Neu3 cells. For each time point analyzed, SCR cells have been used as internal reference samples to define the effects of sialidase-3 up-regulation.

### 2.5. Effects of Sialidase-3 Inhibition on the RISK Pathway Activation

Next, we tested whether inhibition of sialidase-3 would prevent activation of Akt and Erk1/2 kinases during IRI. To this end, OX-Neu3 cells were treated with the specific sialidase Neu3 inhibitor LR332 (50 μM), subjected to IRI in vitro, and then analyzed by Western Blot to determine the activation of both kinases. The results showed that inhibition of sialidase-3 activity significantly reduced Akt (−30 to −60%) and Erk1/2 (−25 to −40%) activation compared with OX-Neu3 cardiomyoblasts exposed to IRI in vitro but untreated with LR332 inhibitor (Figure 7A,B).

### 2.6. RISK Pathway Inhibition Reverts Sialidase-3 Cardioprotection

To test whether inhibition of the RISK pathway could reverse sialidase-3 effects during IRI, H9c2 cells were treated with 50 μM of either LY294002 or PD98059, two specific inhibitors of Akt and Erk1/2 activation, respectively, after optimization of treatment conditions (Appendix A). Akt inhibition by LY294002 was confirmed by western blot analysis in both SCR and OX-Neu3 cells during IRI (Figure 8A). Quantification of WB showed that LY294002 caused a significantly higher Akt decrease in OX-Neu3 myoblasts than in SCR controls particularly during reperfusion. LY294002-untreated cells (either SCR or OX-Neu3 H9c2) exposed to IRI in vitro have been used as internal reference for each time point analyzed (Figure 8B). Analysis of cell growth showed that treatment with LY294002 completely abolished the protective effects mediated by *Neu3* upregulation, as OX-Neu3 myoblasts showed proliferation reduction during IRI that was comparable to that of control cells (Figure 8C). Akt inhibition also caused a 4-fold increase in apoptosis in OX-Neu3 cells during IRI, comparable to that of control cells (Figure 8D). SCR and OX-Neu3 cells were then treated with the Erk1/2 inhibitor PD98059 before exposure to IRI in vitro. Analysis of Erk1/2 activation by WB showed a significant reduction in Erk1/2 phosphorylation in both SCR and even more markedly in OX-Neu3 cells PD98059-untreated cells (either SCR or OX-Neu3 H9c2) exposed to IRI in vitro have been used as internal reference for each time point analyzed (Figure 8E,F). Inhibition of Erk1/2 with PD98059 completely suppressed the cardioprotective effects of sialidase-3 against IRI, as both cell growth and apoptosis of OX-Neu3 cells decreased during IRI similarly to SCR (Figure 8G,H).

## 3. Discussion

Elucidation of the molecular mechanisms of myocardial response to ischemia/reperfusion injury (IRI) is a critical step for the development of new cardioprotective approaches. When exposed to IRI, the myocardium activates endogenous defense mechanisms that could be pharmacologically enhanced. The activation of pro-survival kinases, such as the reperfusion injury salvage kinase (RISK) pathway, may serve as an effective strategy to protect ischemic and reperfused myocardium [29]. Pro-survival kinases are protective when activated acutely, whereas chronic upregulation has been shown to be detrimental due to their growth-promoting effects, resulting in cardiac hypertrophy [30,31]. Consequently, a better understanding of the physiological functioning of the RISK pathway could lead to the development of a more effective approach against IRI. This work shows that sialidase-3, a ubiquitous glycohydrolase present on the cell membrane, modulates the activation of the RISK pathway during IRI, thereby promoting cardiomyocyte survival. These results confirm the highly conserved nature of Neu3 and support previous findings that the enzyme helps regulate the proliferation and survival of skeletal and cardiac myoblasts, particularly during differentiation and in stressful environments [26,32,33,34,35,36]. The current study reports that *Neu3* levels are significantly reduced during the reperfusion phase after myocardial infarction in a mouse model of IRI after LAD ligation. Analysis of *Neu3* expression and activity in an in vitro model of IRI revealed analogous downregulation during reperfusion. However, in the early stages of the ischemic phase, an upregulation of *Neu3* was observed that ultimately sought to promote cardiomyoblast survival. These results are consistent with our earlier study that found transcriptionally upregulated sialidase-3 in the myocardium of cyanotic patients with congenital heart defects, such as Fallot syndrome [36]. Triggered by hypoxia, sialidase-3 activates the hypoxia-inducible factor HIF-1α and its downstream targets, including VEGF, as well as several key genes involved in the glycolytic pathway [37]. The cell protection mechanism mediated by sialidase-3 also appears to function in other cells that rely primarily on aerobic metabolism, such as skeletal muscle [36,38]. Indeed, we have reported that sialidase-3 activates the same anti-apoptotic and pro-survival pathways in skeletal muscle in response to hypoxic stress conditions [26] and is essential for myoblast differentiation [35]. Sialidase-3 has been shown to promote cell resistance by activating the epidermal growth factor receptor (EGFR), which affects its downstream signaling cascades, particularly the Akt and Erk1/2 signaling pathways [26]. Since the major RISK signaling pathways are PI3K-Akt and MEK1-ERK1/2 [20], we tested whether overexpression of *Neu3* would induce cardioprotection. This resulted in a significant increase in proliferation and a decrease in apoptosis when *Neu3* overexpressing cardiomyoblasts were exposed to IRI. As we expected, overexpression of *Neu3* caused activation of both Akt and Erk1/2 kinases during the ischemic phase and at different time points of reperfusion. These effects were completely suppressed by pharmacological inhibition of sialidase-3. Moreover, inhibition of either the Akt or Erk1/2 pathway suppressed cardioprotection in *Neu3*-overexpressing cardiomyoblasts, suggesting that it is mediated by the RISK pathway. In addition, we found that overexpression of *Neu3* caused upregulation of HIF-1α. These results are consistent with our previous findings that sialidase-3 plays a role in activating a HIF-1α-mediated switch of metabolism to glycolysis and that expression of the enzyme correlates positively with HIF-1α levels measured in human cardiac muscle samples [36]. In fact, HIF-1α upregulation has been associated with improved myocardial tolerance to acute IRI due to increased activity of downstream targets, including erythropoietin [39], heme oxygenase-1 (HO-1) [40], and nitric oxide synthase [41]. Stabilization of HIF-1α also triggers the metabolic switch from oxidative phosphorylation to anaerobic glycolysis, reducing mitochondrial ROS production during IRI and counteracting mPTP opening at the onset of myocardial reperfusion [42]. Moreover, overexpression of HIF-1α during myocardial infarction was recently associated with significant downregulation of cardiomyocyte apoptosis by promoting the HO-1-induced antioxidant response [43] and stimulating BNIP3-mediated mitochondrial autophagy, which plays an essential role in cardiac cell protection [44]. However, the role of HIF-1α in cardiac response to stress conditions remains controversial and requires further investigation [45,46,47,48].

In conclusion, our study reveals that sialidase-3 is involved in cardioprotection through stimulation of the RISK pathway and the HIF-1α signaling cascade. Activation of sialidase-3 after myocardial infarction may represent a novel therapeutic target to be explored. To this end, GM3 synthase inhibitors have already demonstrated to mimic sialidase-3 activation pharmacologically [49]. Further studies in this direction are ongoing in our laboratories.

## 4. Materials and Methods

### 4.1. Cell Culture and Treatments

H9c2 rat cardiomyoblasts were obtained from Sigma-Aldrich (St. Louis, MO, USA) and cultured at 37 °C in a 5% CO_2_, 95% air-humidified atmosphere, in Dulbecco’s modified Eagle’s medium with low glucose concentration (1 g/L) (DMEM, Sigma-Aldrich), containing 10% (*v*/*v*) Fetal Bovine Serum (FBS), 2 mM L-glutamine, 100 U/mL penicillin and 100 mg/mL streptomycin (*Growth medium*). To mimic ischemic conditions, cells were cultured for different time lengths in a 5% CO_2_, 1% O_2_ hypoxic work station (SCI-tive, Baker Ruskinn), in the presence of DMEM without glucose, L-glutamine, phenol red, and sodium pyruvate (*Ischemic medium*). Then, cells were shifted back to a 5% CO_2_, 21% O_2_ air-humidified atmosphere and cultured in DMEM with low glucose concentration (1 g/L) and supplements to simulate the reperfusion phase. 

### 4.2. Sialidase Neu3 Stable Overexpression

H9c2 were plated at a density of 1 × 10^5^ cells and transfected with a Rat *Neu3* Lentiviral Vector (pLenti-GIII-CMV-GFP-2A-Puro) (Applied Biological Materials, Richmond, BC, Canada) according to the ViaFect™ Transfection Reagent (Promega Corporation, Madison, WI, USA) manufacturer’s protocol, after reaching 90% confluency. Transfected cells were selected using 10 mg/mL puromycin (Invivogen, San Diego, CA, USA) and the clone with the highest *Neu3* expression and activity was employed for the further experiments. 

### 4.3. RNA Extraction and Quantitative PCR (qPCR)

Total RNA was isolated with Reliaprep^TM^ RNA cell miniprep system (Promega Corporation), following the manufacturer’s protocol, and 1 μg of RNA was reverse transcribed to cDNA using the iScript cDNA synthesis kit (Bio-Rad, Hercules, CA, USA), according to the manufacturer’s instructions. Real time PCR was performed with 10 ng of cDNA template, 0.2 μM primers, and 1 × GoTaq^®^ qPCR Master Mix (Promega Corporation) in 20 μL of final volume, using a StepOnePlus^®^ real time PCR system (Applied Biosystem, Waltham, MA, USA). The amplification protocol was: 95 °C for 2 min, 40 cycles of 5 s each at 95 °C, 30 s at 58 °C and 30 s at 72 °C, and a final stage at 72 °C for 2 min.

The relative quantification of the expression of both mouse/rat target genes was calculated by the equation 2^−ΔΔCt^ using the *Rpl13a* gene as internal housekeeper. Melting curves were monitored to guarantee the accuracy and the specificity of the amplicon. All reactions were performed in triplicate. The primers sequences were obtained using the Primer3 web service and their quality was checked by PCR Primer Stats. The primers sequences are listed in Table 1.

### 4.4. Sialidase Activity Assay

H9c2 cells were collected from 80% confluent dishes by scraping and resuspended in PBS containing a protease inhibitor cocktail (Merck, Kenilworth, NJ, USA). Pulse sonication was used to lyse the cells (10 pulses of 0.5 s in ice) and supernatants were clarified by centrifugation at 800× *g* for 10 min at 4 °C. The membrane fraction was separated by centrifugation at 30,000× *g* for 75 min at 4 °C with an AvantiTM J-30I Centrifuge (Beckman Coulter, Brea, CA, USA). The protein samples concentration was measured by the Pierce BCA Protein Assay kit (Thermo Scientific, Waltham, MA, USA). The sialidase-3 activity associated to this fraction was assayed by incubating 30 µg of protein with the (4-MU-NeuAc) at pH 3.8 for 1 h at 37 °C, and it was read by a multiplate reader (Varioskan Lux, Thermo Scientific) with an excitation wavelength of 365 nm and an emission filter of 448 nm, according to our previously published protocol [50]. One milliunit of sialidase activity is defined as the amount of enzyme liberating 1 nmol of product (4-MU) per min.

### 4.5. Cell Growth Analysis

For all experiments, 1 × 10^5^ H9c2 were plated in 35 mm dishes and exposed to the in vitro model of ischemia/reperfusion. Cells were counted by the trypan blue dye exclusion assay after 12 h of ischemia and 12, 24, 36, and 48 h of reperfusion. Cell number was determined by the automated cell counter Countess II FL™ (Life Technologies, Carlsbad, CA, USA).

### 4.6. Cytotoxicity Detection Test

Cytotoxicity was measured by the CellTox^TM^ Green Cytotoxicity Assay (Promega Corporation), following the manufacturer’s instructions. Briefly, 2.5 × 10^3^ cells were plated in triplicate in 96 dark-walled plates, exposed to 12 h of ischemia and 3, 6, 12, 24, and 48 h of reperfusion. Buffer containing a 1:1000 dilution of CellTox Green Dye was added to each well and incubated at room temperature in the dark for 15 min. Fluorescence data were collected by a multiplate reader (Varioskan Lux, Thermo Scientific) at an excitation wavelength of 480–500 nm and an emission filter of 520–530 nm.

### 4.7. Apoptosis Assay by Hoechst 33342 DNA Staining

7 × 10^4^ cells were plated in 35 mm Petri dishes and exposed to 12 h of ischemia, followed by 3, 6, 24, and 48 h of reperfusion. At any time point analyzed, cells were fixed in paraformaldehyde 4% for 15 min at room temperature (RT) and then washed 3 times with PBS. Blocking and permeabilization were performed in PBS with 5% BSA + 0.1% Triton-X100 for 15 min RT, followed by a 15 min staining with Hoechst 33342 (100 ng/mL in PBS) at RT. Apoptotic cells were analyzed under a fluorescent microscope (Olympus TH4-200, Olympus Corporation, Shinjuku-ku, Tokyo, Japan) with magnification 20×. The % of apoptotic cells was obtained by normalizing the number of altered nuclei by the total number nuclei. The counts have been performed in 15 different fields for each sample. 

### 4.8. Caspase 3/7 Activation Assay

The induction of caspase 3/7 activation was analyzed using the Caspase-Glo^®^ 3/7 Assay (Promega Corporation), according to the manufacturer’s instructions. This kit is based on the caspase-3/7-mediated cleavage of the DEVD sequence of a luminogenic substrate resulting in a luminescent signal, which is proportional to the caspase activity of the samples. Briefly, H9c2 cells were seeded 7 × 10^4^ in 35 mm Petri dishes and exposed to 1, 3, 6 or 12 h of ischemia and 3, 6, 12, 24, and 48 h of reperfusion. At the end of each time point, 350 µL of Caspase-Glo^®^ 3/7 Reagent were added directly to the samples and incubated 1 h at RT. At the end of the incubation, 200 µL of each sample were transferred into a 96 white-walled plate in triplicate and the luminescent signal was measured using a plate-reading luminometer (Varioskan Lux, Thermo Scientific). For each time point analyzed, a cell count was performed by the trypan blue dye exclusion assay, as already described, to normalize the levels of caspase 3/7 activation.

### 4.9. Protein Extraction and Western Blot Analysis

For protein expression analysis, cells lysates were prepared following a well-established protocol, as previously described [27]. The protein samples concentration was measured by the Pierce BCA Protein Assay kit (Thermo Scientific). Then, 30 µg of total protein were separated by SDS-Page, before being transferred to nitrocellulose membranes. To block non-specific binding sites, membranes were incubated with Tris-HCl buffer pH 7.5 (TBS) containing 5% bovine serum albumin (BSA) or 5% non-fat dried milk for 1 h at RT and then incubated at 4 °C overnight with the following primary antibodies: anti-phospho-Akt T308 (1:1000, Cell signaling), anti-Akt (1:1000, Cell signaling), anti-phospho-p44/42 MAPK (p-Erk1/2) T202/Y204 (1:1000, Cell signaling), anti-p44/42 MAPK (Erk1/2) (1:2000, Cell signaling), anti-HIF-1α (1:1000, Cell signaling), anti-Calnexin (1:10,000, Abcam). The membranes were washed three times with TBS-Tween 20 for 10 min and then incubated at RT for 2 h with the appropriate HRP-conjugated secondary antibodies. Membranes were subsequently washed with TBS-Tween 20 and proteins were detected using an ECL detection kit (Cyanagen, Bologna, Italy), according to the manufacture’s protocol. Membranes’ acquisition was performed with the LI-COR Odissey Infrared Imaging System and bands intensity was quantified using the Image Studio Lite software (LI-COR Biotechnology, Lincoln, NE, USA). All of the bands analyzed corresponded to the expected molecular weights and their intensity has been normalized by the intensity of the calnexin protein, which was used as housekeeper.

### 4.10. Ganglioside GM3 Content Analysis

The determination of the ganglioside GM3 content in control and *Neu3*-overexpressing H9c2 cells was performed by sphingolipids radioactive labeling, as previously described [49]. Briefly, [3-^3^H]sphingosine (PerkinElmer Life Sciences) was dissolved in growth medium to obtain a final sphingosine concentration of 0.25 μCi/per 100-mm dish and 1 × 10^6^ cells were incubated with this medium for 2-h. Then, the medium was removed and the cells were chased for 48 h with normal culture medium (10% FBS DMEM) to reach the metabolic steady state. At the end of chase, the cells were harvested, frozen and lyophilized for radioactive lipids extraction. Total lipids were extracted twice with chloroform/methanol 2:1 (*v*/*v*) and with chloroform/methanol/water 20:10:1 (*v*/*v*/*v*), respectively. The resulting lipid extracts were subjected to a two-phase portioning and the obtained aqueous phase, containing gangliosides, was separated by high-performance thin layer chromatography (HPTLC), using the solvent system chloroform/methanol/0.2% aqueous CaCl_2_, 60:40:9 (*v*/*v*/*v*). 

Radioactive lipids were visualized with a Beta-Imager 2000 (Biospace, Paris, France) and identified by comparison with radiolabeled standards. The radioactivity associated with the individual lipids was quantified with the specific β-Vision software (Biospace, Paris, France) and normalized by the protein total content (mg) of each sample.

### 4.11. Sialidase NEU3 Activity Inhibition

To inhibit the sialidase-3 activity, *Neu3*-overexpressing H9c2 cells were treated with increasing concentrations (10–50–100 μM) of the newly characterized sialidase Neu3 specific inhibitor C9-modified Zanamivir analogue (LR332), synthesized in our laboratory. After 12 h of incubation with LR332, *Neu3*-overexpressing H9c2 were collected and tested for sialidase-3 activity. The selected concentration of the inhibitor was 50 μM, which was used in the further experiments and applied during the entire cells’ exposure to the IRI in vitro model.

### 4.12. RISK Pathway Inhibition

To inhibit the activation of Akt and Erk1/2, two of the major kinases involved in the RISK pathway, SCR and OX-Neu3 H9c2 cells were treated with two specific commercially available inhibitors. In particular, LY294002 (LY; Cell Signaling Technology, Danvers, MA, USA), which is a potent cell permeable inhibitor of the phosphatidylinositol 3-kinase (PI3K), is able to block the PI3K-dependent Akt phosphorylation. PD98059 (PD; Cell Signaling Technology), which is instead a highly selective inhibitor of the Mek1 activation and of the MAP kinase cascade, was used to block Erk phosphorylation on both threonine and tyrosine residues. Both inhibitors were used at a 50 μM concentration. OX-Neu3 and SCR cells were incubated with both inhibitors 1 h before being exposed to IRI and during the entire duration of the experiments.

### 4.13. In Vivo Experiments

Animal studies were performed according to the animal protocol guidelines described by the Institutional Animal Care and Use Committee (IACUC) authorization number 388/2018-PR at San Raffaele Scientific Institute (Milan, Italy). All mice were housed for two weeks in individual cages with a 12-h light/dark cycle, allowing free access to food and water. All efforts were made to minimize animal suffering and to reduce the number of mice used, in accordance with the European Communities Council Directive of November 24, 1986 (86/609/EEC).

### 4.14. Left Anterior Descending (LAD) Coronary Artery Ligation

For the in vivo experiments, 25 male C57BL/6N mice (8–10 weeks old; Charles River Laboratories Italia, Italy) were subjected to LAD coronary occlusion, as previously described [14,51]. Briefly, animals were anesthetized by intraperitoneal injection of Medetomidine, 0.5 mg/Kg (Orion Pharma S.r.l., Milano, Italy) and Ketamine, 100 mg/Kg (Merial, Ingelheim am Rhein, Germany). When completely unconscious, the mouse’s chest was opened between the second and the third rib to expose the left ventricle. Once recognized and located the LAD, a silk suture was passed under the coronary vessel, and a 5-mm long piece of tubing was placed. Then, the knot was tightened around the artery and tubing, simulating the ischemic phase. After 30 min of ischemia, coronary occlusion was released and the heart reperfused. The animals were subjected to 30 min of ischemia or 1, 4, and 7 days of reperfusion. Sham animals were subjected to anesthesia, chest opening without coronary occlusion, and sacrificed on the same day of surgery. Each experimental group was composed of 5 animals. 

### 4.15. Evan’s Blue/TTC Double Staining

To discriminate the infarct area from the healthy cardiac tissue, a double staining with Evan’s blue and triphenyl tetrazolium chloride (TTC) was performed. At the end of each time point, the coronary artery was re-occluded, 200 μL of 4% (*w*/*v*) Evans blue solution was injected into the apex of the left ventricle to identify the ischemic area by dye exclusion, and the heart was excised and frozen in liquid N_2_ to be stored at −80 °C until analysis. Then, each heart was cut into 1 mm thick slices and incubated with 1% TTC solution for 5 min in order to identify the area at risk (AAR). At this point, both the infarct and the AAR areas were collected together and separated from the healthy cardiac tissue for additional biochemical analyses. 

### 4.16. Echocardiography

Cardiac functionality was measured in terms of ejection fraction (EF) and fractional shortening (FS) by echocardiography, as previously reported [52].

### 4.17. Statistical Analysis

All assays were performed from three up to six replicates, and the quantitative data are displayed as box and whisker plots. Each square present in the graphs represent a single replicate. The nonparametric Mann-Whitney test or Kruskall-Wallis test were used to determine statistical significance using GraphPad Prism 9 software, depending on the type of data (See figures’ captions). *p* values of less than 0.05 were considered to be significant.

## 5. Study Limitations

The results of this study are based on in vitro experiments with a commercial rat heart-derived cell line (H9c2) that is characterized by a permanent division ability but still retains some cardiomyocyte characteristics. Although the use of H9c2 cells has been recognized as a valid and suitable in vitro model for simulating various cardiac pathological processes, including ischemia and reperfusion injury [53,54,55], they cannot perfectly recapitulate the characteristics of primary cardiomyocyte cultures. Moreover, the involvement of sialidase-3 in the cardiac response to IRI in vivo has been studied only in terms of mRNA expression. The possibility of investigating how IRI exposure modulates sialidase-3 protein expression and cellular localization in vivo would provide valuable information on the pathophysiological role of this enzyme. However, it is known that none of the commercially available antibodies against sialidase-3 are accurate and specific for this protein, making it unsuitable for immunohistochemical techniques.

## Figures and Tables

**Figure 1 ijms-23-06090-f001:**
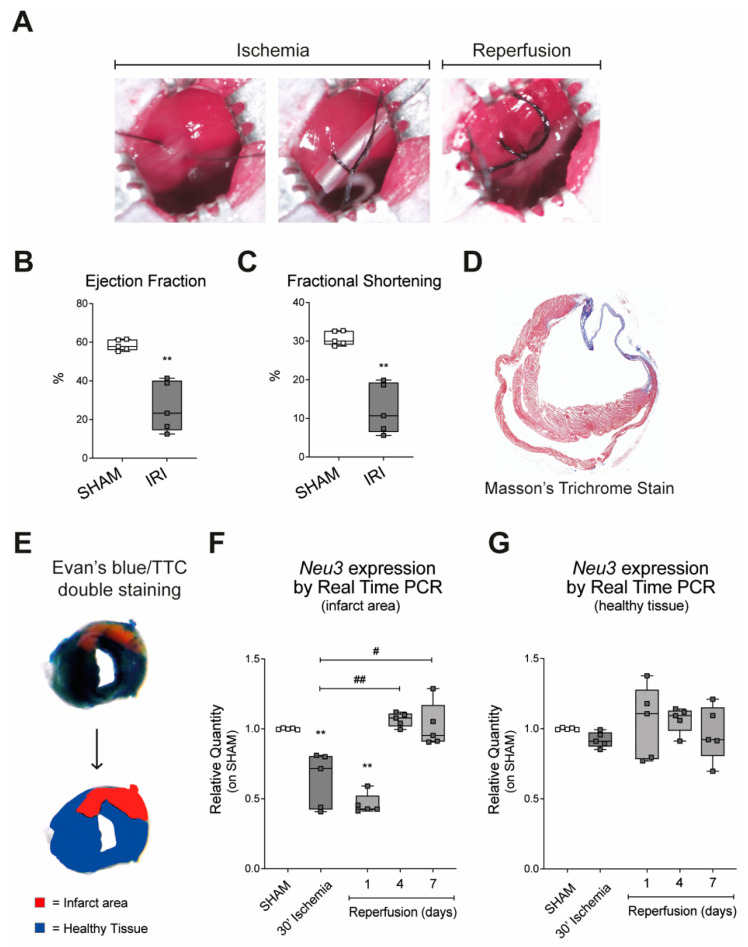
**Effects of ischemia and reperfusion exposure on the expression of sialidase *Neu3*.** (**A**) Ischemia and reperfusion injury in vivo model obtained by temporary ligation of the left anterior descending coronary vessel; (**B**,**C**) cardiac function was assessed by echocardiography. Ejection fraction (**B**) and fractional shortening (**C**) are shown as percentages, in comparison between sham and animals subjected to IRI. (**D**) Morphological cardiac analysis to assess fibrotic scarring with Masson’s trichrome staining; (**E**) Evan’s blue/TTC double staining schematic representation; (**F**) sialidase *Neu3* expression in the infarcted area measured by Real Time PCR (**G**) sialidase *Neu3* expression in healthy cardiac tissue measured by Real Time PCR. Data are expressed as relative amounts compared with sham animals. Each square in the graphs represent a single animal. Statistical significance was determined by the nonparametric Kruskal-Wallis test.; ** *p* < 0.01. The # symbol indicates the statistical significance as compared to 30′ ischemia: # *p* < 0.05; ## *p* < 0.01.

**Figure 2 ijms-23-06090-f002:**
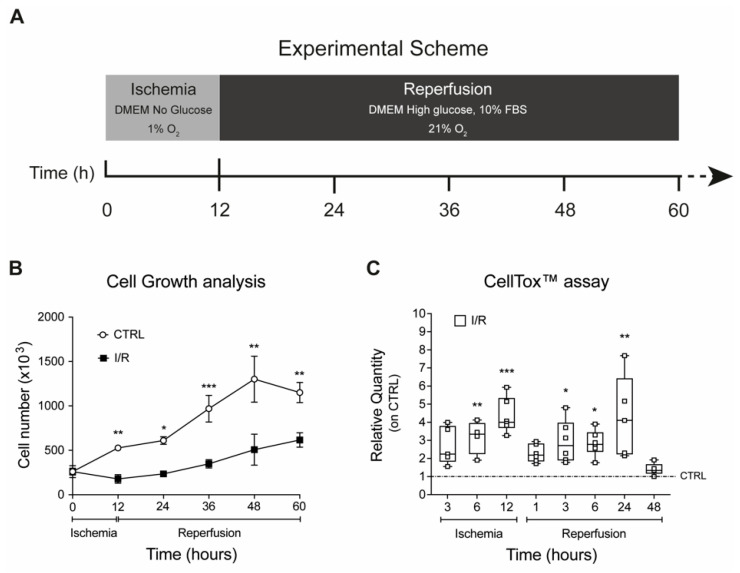
**Effects on cell proliferation and cytotoxicity in an in vitro model of IRI.** (**A**) Schematic representation of in vitro model of IRI; effects of IRI on H9c2 in terms of proliferation (**B**) and cytotoxicity (**C**). Each square in the graphs represent an experimental replicate. In the legend: CTRL represents H9c2 cells maintained in normoxic culture conditions; I/R represents H9c2 cells exposed to ischemia and reperfusion. Statistical significance was determined by the nonparametric Kruskal-Wallis test. * *p* < 0.05; ** *p* < 0.01; *** *p* < 0.001.

**Figure 3 ijms-23-06090-f003:**
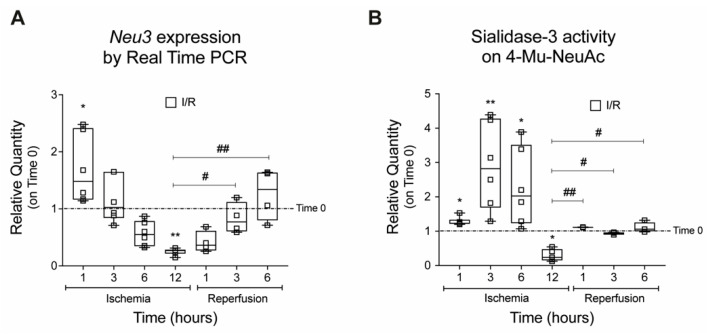
**Effects of ischemia and reperfusion on *Neu3* expression and Sialidase-3 activity**. (**A**) Expression of sialidase *Neu3* measured by real time PCR; (**B**) enzymatic activity of sialidase-3 measured on the synthetic substrate 4-Mu-NeuAc. Data are expressed as relative quantities compared with H9c2 not exposed to IRI and collected at the beginning of the experiments (*Time 0 dashed line*). Each square in the graphs represent an experimental replicate. Statistical significance was determined by the nonparametric Kruskal-Wallis test. * *p* < 0.05; ** *p* < 0.01. The # symbol indicates the statistical significance as compared to 30′ ischemia. # *p* < 0.05; ## *p* < 0.01.

**Figure 4 ijms-23-06090-f004:**
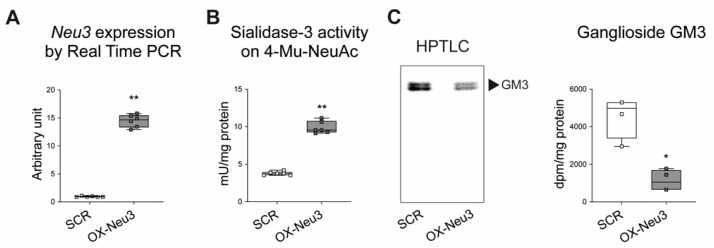
**Sialidase-3 overexpression in H9c2 cardiomyoblasts**. (**A**) Expression of sialidase *Neu3* measured by Real Time PCR; (**B**) enzymatic activity of sialidase-3 measured on the synthetic substrate 4-Mu-NeuAc; (**C**) analysis of ganglioside GM3 by labeling with [3-^3^H]-sphingosine. Left panel (**C**): radiochromatographic image of HPTLC separation of gangliosides contained in the aqueous phase. Right panel (**C**): GM3 quantification expressed as dpm/mg protein. Each square in the graphs represent an experimental replicate. Statistical significance was determined by the nonparametric Mann-Whitney test. * *p* < 0.05; ** *p* < 0.01.

**Figure 5 ijms-23-06090-f005:**
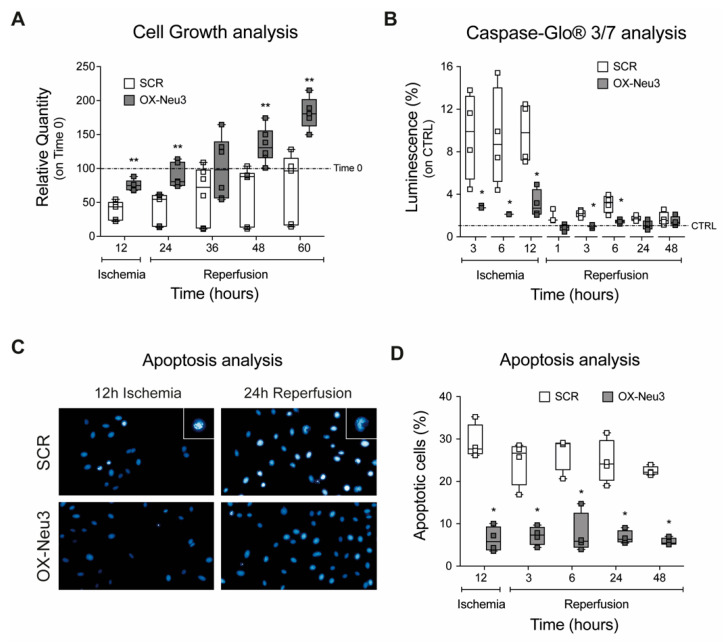
**Effects of *Neu3* overexpression in H9c2 cells exposed to IRI.** (**A**) Cell growth analysis. Data are expressed as relative quantity (%) as compared to both SCR or OX-Neu3 cells at the beginning of IRI treatment (*Time 0 dashed line*) (**B**) caspase 3/7 activation analysis through the Caspase-Glo^®^ 3/7 assay. Data are expressed as fold change as compared to both SCR or OX-Neu3 cells maintained in standard culture conditions and not exposed to IRI, used as controls (*CTRL dashed line*); (**C**) apoptotic nuclei staining by Hoechst 33,342; White squares represent apoptotic cells magnification; (**D**) apoptosis level analysis expressed as percentage of apoptotic nuclei per total nuclei. Each square in the graphs represent an experimental replicate. Statistical significance was determined by the nonparametric Kruskal-Wallis test by comparing OX-Neu3 cells to SCR cells at any time point analyzed. * *p* < 0.05; ** *p* < 0.01.

**Figure 6 ijms-23-06090-f006:**
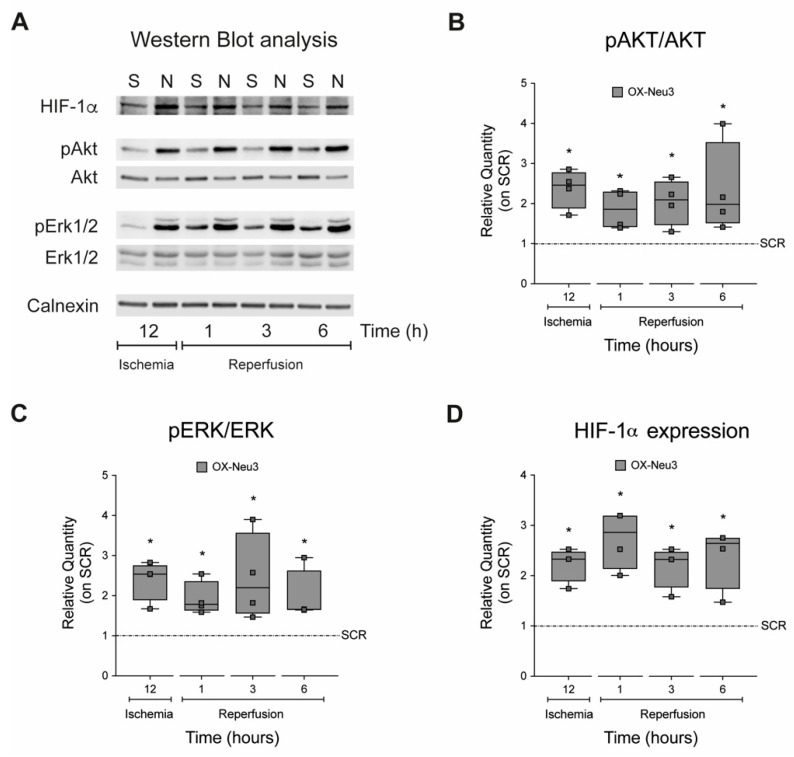
**Effects of sialidase *Neu3* overexpression on the RISK pathway activation and HIF-****α expression.** Activation of the RISK pathway and expression of HIF-1α in scramble and OX-Neu3 cells exposed to IRI were evaluated by Western Blot. (**A**) Western Blot of HIF-1α; phospho-Akt (Thr308) and total Akt; phospho-Erk1/2 (Thr202/Tyr204) and total Erk1/2; Calnexin; (**B**) ratio between phospho-Akt (Thr308) and total Akt; (**C**) ratio between phosho-Erk1/2 (Thr202/Tyr204) and total Erk1/2; (**D**) HIF-1α expression level. Calnexin was used as a housekeeper to normalize protein expression levels. “S” is the abbreviation for SCR cells; “N” is the abbreviation for OX-Neu3 cells. Data are expressed as relative amounts compared with scramble cells, used as internal control for each time point analyzed (*SCR dashed line*). Each square in the graphs represent an experimental replicate. Statistical significance was determined by the nonparametric Kruskal-Wallis test by comparing OX-Neu3 cells to SCR cells at any time point analyzed. * *p* < 0.05.

**Figure 7 ijms-23-06090-f007:**
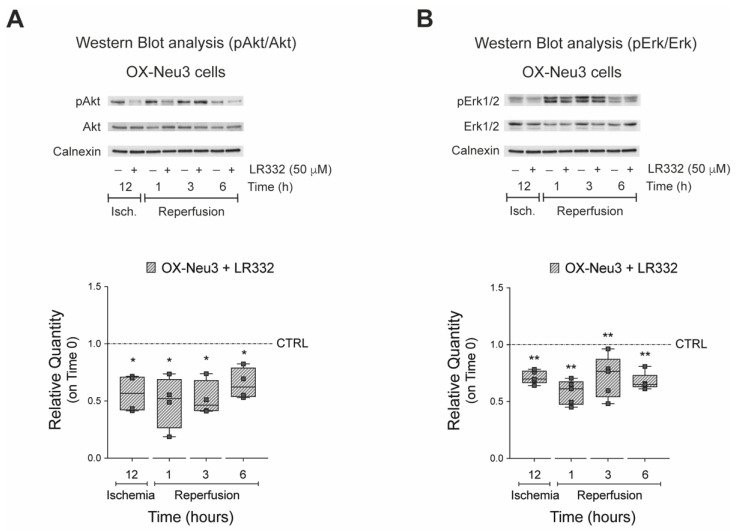
**Effects of sialidase-3 inhibition on activation of the RISK pathway**. Activation of Akt and Erk1/2 in OX-Neu3 H9c2 treated with LR332 (arbitrary name given to the non-commercially available sialidase3 inhibitor synthesized by our group) was examined by WB. (**A**) WB analysis of phospho-Akt (Thr308) and total-Akt (upper panel); ratio between phospho-Akt (Thr308) and total-Akt (lower panel); (**B**) WB analysis of phospho-Erk1/2 (Thr202/Tyr204) and total-Erk1/2 (upper panel); ratio between phospho-Erk1/2 (Thr202/Tyr204) and total-Erk1/2 (lower panel). Data are expressed as relative amounts compared with LR332 untreated OX-Neu3 cells exposed to IRI in vitro (*CTRL dashed line*). Each square in the graphs represent an experimental replicate. Statistical significance was determined by the nonparametric Kruskal-Wallis test. * *p* < 0.05; ** *p* < 0.01.

**Figure 8 ijms-23-06090-f008:**
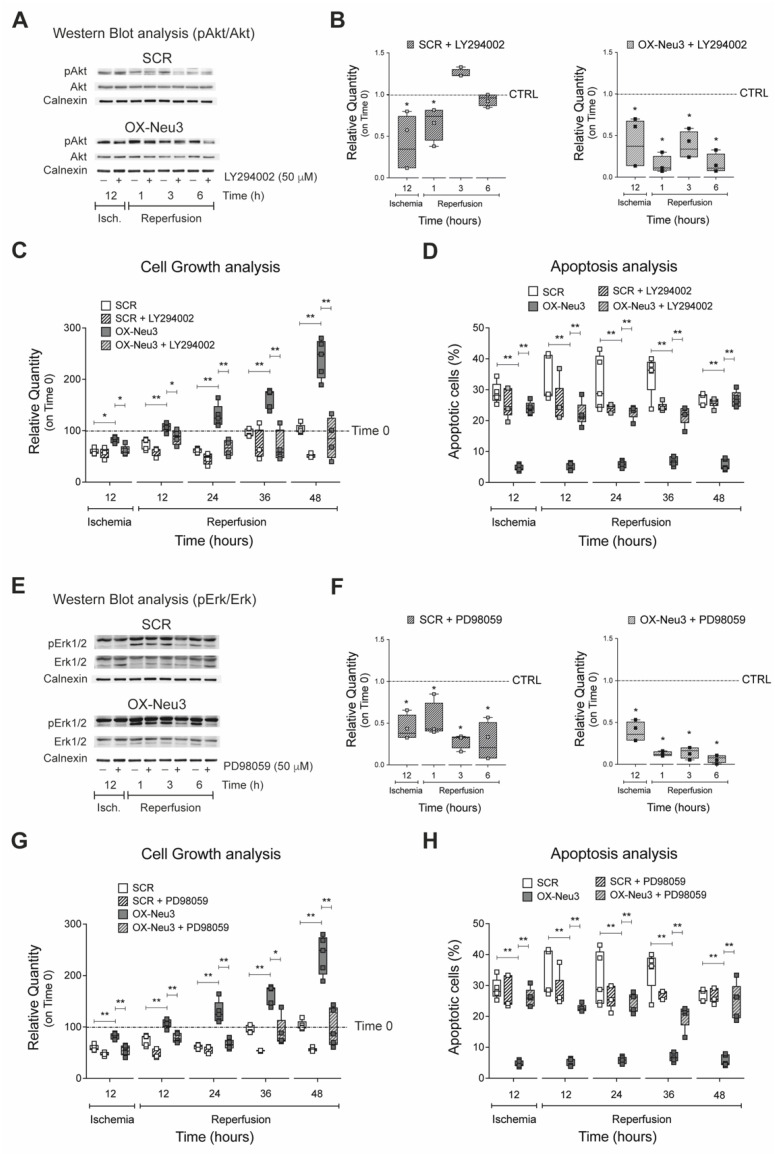
**Effects of the RISK pathway inhibition on the sialidase-3 mediated cardioprotective effects.** (**A**) WB analysis of phospho-Akt (Thr308) and total-Akt in scramble and OX-Neu3 cells treated with LY294002. Calnexin was used as a housekeeper to normalize protein expression; (**B**) ratio between phospho-Akt (Thr308) and total-Akt. The levels of Akt activation, at each time point analyzed, have been expressed as relative quantity using either LY294002-untreated SCR or OX-Neu3 cells exposed to IRI in vitro as internal reference control (*CTRL dashed line*); (**C**) analysis of cell growth of scramble and OX-Neu3 cells treated with LY294002 and exposed to IRI. Data are expressed as relative quantity (%) compared to either SCR or OX-Neu3 cells at the beginning of IRI exposure (*Time 0 dashed line*); (**D**) analysis of apoptosis level of scramble and OX-Neu3 cells treated with LY294002 and exposed to IRI. Results are expressed as percentage of apoptotic nuclei per total nuclei; (**E**) WB analysis of phospho-Erk1/2 (Thr202/Tyr204) and total Erk1/2 in both scramble and OX-Neu3 cells treated with PD98059. Calnexin was used as a housekeeper to normalize protein expression; (**F**) ratio between phospho-Erk1/2 (Thr202/Tyr204) and total Erk1/2. The levels of Erk1/2 activation, at each time point analyzed, have been expressed as relative quantity using either PD98059-untreated SCR or OX-Neu3 cells exposed to IRI in vitro as internal reference control (*CTRL dashed line*); (**G**) cell growth analysis of scramble and OX-Neu3 cells treated with PD98059 and exposed to IRI. Data are expressed as relative quantity (%) compared to either SCR or OX-Neu3 cells at the beginning of IRI exposure (*Time 0 dashed line*); (**H**) apoptosis analysis of scramble and OX-Neu3 cells treated with PD98059 and exposed to IRI. Results are expressed as percentage of apoptotic nuclei per total nuclei. Each square in the graphs represent an experimental replicate. Statistical significance was determined by the nonparametric Kruskal-Wallis test. * *p* < 0.05; ** *p* < 0.01.

**Table 1 ijms-23-06090-t001:** qPCR primers sequences.

Gene	Forward Primer	Reverse Primer
Rat *Neu3*	5′-ATGCCCTCTGATGGACAGAT-3′	5′-CATGTCCCTGATGGTGCTC-3′
Rat *Rpl13a*	5′-TCTCCGAAAGCGGATGAACAC-3′	5′-CAACACCTTGAGGCGTTCCA-3′
Mouse *Neu3*	5′-TGCGTGTTCAGTCAAGCC-3′	5′-GCAGTAGAGCACAGGGTTAC-3′
Mouse *Rpl13a*	5′-CTCGGCCGTTCCTGTAT-3′	5′-GTGGAAGTGGGGCTTCAGTA-3′

## Data Availability

The raw data supporting the conclusions of this manuscript will be made available by the authors, without undue reservation, to any qualified researcher.

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
