# Peer review of "Neu3 Sialidase Activates the RISK Cardioprotective Signaling Pathway during Ischemia and Reperfusion Injury (IRI)"

_ijms, 2022, doi:10.3390/ijms23116090_

Round 1

Reviewer 1 Report

The current study builds on previous publications by this group investigating the role of Neu3 sialidase in cardiac ischemic injury. They authors have a solid foundation in the subject, and seek to reveal mechanisms through which Neu3 may confer protection following IRI at the tissue and cellular level. The study begins with an in vivo mouse model for IRI, then transitions directly to an in vitro IRI model using H9c2 cells as a proxy for cardiomyocytes. The introduction describes the scientific background and questions well, methods are clear although minor clarification of reference controls in western blots is needed. The manuscript is well written, data are well presented, and conclusions are supported by the results. Comments are provided to strengthen the manuscript for publication in IJMS.

Figure 1: The in vivo data give a clear indication of Neu3 expression following IRI. Localization of Neu3 in treated sections would provide valuable information regarding which cells express Neu3 at baseline, which cells upregulate Neu3 in response to IRI, and what happens to the cellular localization of Neu3 in these cells. If an adequate antibody is not available, these questions could be addressed in the discussion. 

H9c2 cells are useful for performing cell proliferation assays, however they do not represent functional cardiomyocytes. Neonatal ventricular cardiomoyctes (NRVMs) from rats or mice, or isolated adult cardiomyocytes better reflect how Neu3 expression and localization change in response to in vitro IRI conditions. NRVMs work particularly well for signaling experiments. At the very least, include this concern in a paragraph addressing limitations of the study.

Figure 5C and Figure 6D should read "apoptosis levels expressed as percentage of apoptotic nuclei per total nuclei."

Figure 7. The western blots are fine, but it's not clear exactly what the SCR control line represents in B-D. Are levels being compared to cells transduced with control (SCR) lentivirus cultured under standard culture conditions (i.e. no IRI)? Or is each treatment group being compared to its internal SCR control? 

Figure 8. Similar question about blot quantification: Are samples being compared to Neu3 expressing cells not exposed to inhibitor and not exposed to IRI? Because there are multiple treatments happening simultaneously, it's important to clarify what control means, especially since control samples are not shown in the blots.

Figure 9. Same question as for 7 and 8.

Include a section addressing limitations of the study.

Reviewer 2 Report

The study of pathophysiological mechanisms leading to the development of the diseases, as well as the study of ways to influence on these mechanisms to prevent the development of pathological conditions, is very important. The authors describe the modeling of ischemia and reperfusion injury, which are inextricably linked with the restoration of blood flow after acute myocardial infarction. This problem is very significant, because mortality from cardiovascular diseases remains the main cause of population decline in developed countries.

I have a number of questions and comments  which are represented by points.

  1. Statistical Data Analysis.

In paragraph 4.17 the authors indicate that Statistical significance was determined by the nonparametric Mann-Whitney test. Why did the authors present information as mean and standard deviation? Non-parametric statistics include median and quartiles (the first and third quartiles).

In addition, it is more convenient to use the Kruskal-Wallis test (for unrelated samples) or Friedman test (if the samples are related). The Kruskal-Wallis test is used for comparing two or more independent samples of equal or different sample sizes. Correction for multiple comparisons in this case should be put.

When the authors measured the cytotoxicity of cells at different time points, they used the same well, which they measured during ischemia, and then it was measured during reperfusion, if I understand correctly. In this case, it is appropriate to use the Friedman test (for samples greater than two) or the Wilcoxon test (for two samples). The authors should double-check this information in the article.

In the materials and methods (and in the captions to the figures) it is recommended to indicate for which experiments which statistical processing was used.

I recommend presenting the data in the form of a box and whiskers plot with the parameter «min to max. Show all points». The Boxplot has 5 characteristics: the minimum and the maximum (whisker borders), the sample median (located in the middle of the box), and the first and third quartiles (box borders). The replicates are represented by dots on the graph. The dots can be marked with color or different shapes. Boxplot visually characterizes the entire sample as a whole.

  1. Paragraph 2.1. The authors characterize the expression of the Neu3 gene (the mouse gene should be written as follows (https://www.ncbi.nlm.nih.gov/gene)). The names of the genes are written in italics, the size of the letters depends on the organism under study. The authors indicated in table 1 primers only for rats. Are the primers designed for rats valid for use in mice? It is necessary to add comments to paragraph 4.3.

The Neu3 gene in rats and mice (NEU3 in humans) encodes a protein called Sialidase-3 (https://www.uniprot.org/uniprot/?query=Sialidase-3&sort=score). The Authors are encouraged to use this information to edit the places where a gene or protein is mentioned in the text.

  1. Figure 1. Indicate in the caption to the figure information about the meaning of the squares on the graphs. Does the number of squares in the group correspond to the number of animals? Add explanation for «****»
  2. Figure 1 (E). What does the «grid» symbol mean? Add an explanation to the signature.
  3. Figure 2 (С). Indicate in the caption to the figure information about the meaning of the squares on the graphs. Does the number of squares in the group correspond to the number of replicates? Add explanation for CTRL (cells cultured under normoxic conditions), I/R (cells induced by ischemia and reperfusion).
  4. Figure 3. You should correct the spelling of gene and protein and add an explanation to the sign «grid». You should clarify that the number of squares corresponds to the number of replicates per group. Can the authors explain the reason for the different number of measurements at time points? So, for example, we take figure 3 (A). At point zero, the number of measurements is not clear. We have 5 measurements In the ischemia group (1 hour), in the ischemia group (3-12 hours) - 6 measurements. In the reperfusion group we have 4 measurements. Why? Add explanation for «****».
  5. Figure 3 (В). You need to specify the units of measure. Check in the text of the article whether the units of measurement are indicated everywhere on the graphs.
  6. Line 138. The authors are encouraged to designate transduced H9c2 cells with a different name from NEU3. This will avoid confusion because the NEU3 gene codes for Sialidase-3 in humans. This new designation should be entered in the text, in figure 4 and in the caption to it. This also applies to figures S2, 5, 6, S3, 7, 8, 9.
  7. Line 139. «….with a scrambled lentiviral vector and used as a control.» Add explanation for SCR .
  8. Figure In the caption to the figure, when you describe a gene, you should use «Neu3» and when you describe a protein you should use «Sialidase-3». Add explanation for SCR and «****»
  9. Figure 5 (С). Indicate in the caption to the figure the designation of the image in the upper right corner in the SCR group.
  10. Figure 5. Indicate in the caption to the figure information about the meaning of the squares on the graphs. Does the number of squares in the group correspond to the number of replicates? Add explanation for «****»,CTRL и
  11. Figure 6. When you describe a gene, you should use «Neu3», and when you describe a protein - «Sialidase-3». You should edit it in the caption to the figure and in the figure. Indicate in the caption to the figure information about the meaning of the squares on the graphs. Does the number of squares in the group correspond to the number of replicates? Add explanation for «****».
  12. Line 205. You should decrypt the designation LR322 in the caption to the figure.
  13. Figure 7 (А). You should decrypt the designation «S» and «N». Indicate in the caption to the figure information about the meaning of the squares on the graphs. I believe that your addition of the Gapdh/Histone H3 immunoblot will improve your article (see comments to Paragraph 4.9). Add explanation for «****» and SCR.
  14. Figure 8. Comments as to figure 5. Add explanation for «CTRL».
  15. Figure S The authors should bring protein designations to a common form. You need to replace «Clx» with «Calnexin» (as in the main body of the article). I believe that your addition of the Gapdh/Histone H3 immunoblot will improve your article (see comments to Paragraph 4.9). It is necessary to explain what the lower panel means in letters (A) and (B).
  16. Figure 9. Comments as to figure 5. I believe that your addition of the Gapdh/Histone H3 immunoblot will improve your article (see comments to Paragraph 4.9).
  17. Line 372. Why did the authors choose 21% O2 air-humidified atmosphere? The authors did not take the numbers 25% or 50%, did they?
  18. Paragraph I think that when you describe cell experiments you should add Cell Confluence Percentage Value. For example, cells that fill the culture dish by 50% will collect less protein and RNA than cells with 90-95% confluence.
  19. Paragraph 4.3. In addition, the authors should note whether they developed the primers themselves or was it ready-made. If the authors developed on their own, they should note what programs were used to develop and check the quality of primers (Primer-BLAST, PCR Primer Stats, Multiple Primer Analyzer)? After that authors can already refer to table 1, where primer sequences are indicated.
  20. Line 394. The gene in this line has not been fully described. The object of the study is a rat/mouse. Therefore, the genes were Rpl13a (RPL13A in humans) and Neu3 (NEU3 in humans). It is necessary to make these changes throughout the text (including figures) and in the Table 1. This applies to everything where the authors describe genes.
  21. The authors did not use the another housekeeper as an internal control, did they? Why? It could be Gapdh, B2m or Actb.
  22. Paragraph 4.4. и9. It is necessary to add information about the kit with which the authors measured the amount of collected protein.
  23. Line 484. The Authors should decrypt the abbreviation HPTLC (High-performance thin-layer chromatography).
  24. Paragraph 4.5. I think that the authors had removed the cells from the cell dish before counting the cells on the Countess II…. What did the authors do with the cells after counting? The paragraph does not mention this.
  25. Paragraph 4.9. Is it possible for the authors to make a Western Blot using another loading control? The intensity of the Calnexin band in the samples is variable. The Authors can use, for example, Mitochondria Fraction Western Blot Cocktail (ab139416) as loading control (it contains Gapdh/histone H3).
  26. Paragraph 4.11. What time frame was used when you adjusted the concentration of LR322? If this is indicated, then figure 6 (A) becomes clear.
  27. Paragraph 4.12. I recommend to add a manufacturer for LY294002 and PD98059.
  28. Paragraph 4.14. It is necessary to indicate the total number of animals that were used in the experiment. There were 25 of them, right? In brackets, as an explanation, you can specify the number of animals per group. It is still not entirely clear to me how the animals were withdrawn from the experiment…. Five mice of the SHAM group were withdrawned after surgery after 30 minutes (in parallel with the ischemia group), right? The SHАM group was not used after 1, 4, and 7 days, right? It could control for these time points.
  29. Paragraph 4.15. The authors here described a technique for dividing the infarct area from the healthy cardiac tissue. Is it possible for the authors to insert an explanatory photograph in figure 1 (A)? In this case the origin of healthy tissue in Figure 1 (F) becomes clear.

The study is very interesting and can be published after revision of these points.

Author Response

We are sorry we did not address these comments, as they were not sent on the decision letter dated April 26 which requested our reply within 10 days. We feel that these comments probably arrived late and were not taken into consideration by the Editor in his decision as they appeared on the web while we were uploading our revised version based on the Editor's requests and deadline. 

Reviewer 3 Report

The MS is of good quality and clearly written and good setup of experiments as well. However, I have some comments:

1) Effects on cell proliferation, toxicity, and Neu3 modulation of an in vitro model of IRI-----> I think is not appropriate to describe it as a MODULATION,  you just describe an expression level of NEU in an in vitro model of IRI,  I would suggest rephrasing this title. 

2) Clone 3 was selected for further experiments because it had the highest sialidase 140 NEU expression ----> I think should be appropriate also to have a blot to show the protein level. 

3) Why did the authors decide to use H9c2 myoblasts? H9c2 is isolated from embryonic rat tissue I think should be more appropriate to use  HL-1 Cardiac Muscle Cell Line or set up the same set of experiments to confirm the data.

4) Fig 6 is redundant, I think is not necessary to add this set up of experiments, which, indeed, is already described in the final paragraph where you described the mechanism and the pathway as well (RISK Pathway Inhibition reverts Neu3 cardioprotection).

Round 2

Reviewer 2 Report

There was a system bug in the first round of peer review. I think so...
I resubmit my comments in the second round of review.

The study of pathophysiological mechanisms leading to the development of the diseases, as well as the study of ways to influence on these mechanisms to prevent the development of pathological conditions, is very important. The authors describe the modeling of ischemia and reperfusion injury, which are inextricably linked with the restoration of blood flow after acute myocardial infarction. This problem is very significant, because mortality from cardiovascular diseases remains the main cause of population decline in developed countries.

I have a number of questions and comments  which are represented by points.

  1. Statistical Data Analysis.

In paragraph 4.17 the authors indicate that Statistical significance was determined by the nonparametric Mann-Whitney test. Why did the authors present information as mean and standard deviation? Non-parametric statistics include median and quartiles (the first and third quartiles).

In addition, it is more convenient to use the Kruskal-Wallis test (for unrelated samples) or Friedman test (if the samples are related). The Kruskal-Wallis test is used for comparing two or more independent samples of equal or different sample sizes. Correction for multiple comparisons in this case should be put.

When the authors measured the cytotoxicity of cells at different time points, they used the same well, which they measured during ischemia, and then it was measured during reperfusion, if I understand correctly. In this case, it is appropriate to use the Friedman test (for samples greater than two) or the Wilcoxon test (for two samples). The authors should double-check this information in the article.

In the materials and methods (and in the captions to the figures) it is recommended to indicate for which experiments which statistical processing was used.

I recommend presenting the data in the form of a box and whiskers plot with the parameter «min to max. Show all points». The Boxplot has 5 characteristics: the minimum and the maximum (whisker borders), the sample median (located in the middle of the box), and the first and third quartiles (box borders). The replicates are represented by dots on the graph. The dots can be marked with color or different shapes. Boxplot visually characterizes the entire sample as a whole.

  1. Paragraph 2.1. The authors characterize the expression of the Neu3 gene (the mouse gene should be written as follows (https://www.ncbi.nlm.nih.gov/gene)). The names of the genes are written in italics, the size of the letters depends on the organism under study. The authors indicated in table 1 primers only for rats. Are the primers designed for rats valid for use in mice? It is necessary to add comments to paragraph 4.3.

The Neu3 gene in rats and mice (NEU3 in humans) encodes a protein called Sialidase-3 (https://www.uniprot.org/uniprot/?query=Sialidase-3&sort=score). The Authors are encouraged to use this information to edit the places where a gene or protein is mentioned in the text.

  1. Figure 1. Indicate in the caption to the figure information about the meaning of the squares on the graphs. Does the number of squares in the group correspond to the number of animals? Add explanation for «****»
  2. Figure 1 (E). What does the «grid» symbol mean? Add an explanation to the signature.
  3. Figure 2 (С). Indicate in the caption to the figure information about the meaning of the squares on the graphs. Does the number of squares in the group correspond to the number of replicates? Add explanation for CTRL (cells cultured under normoxic conditions), I/R (cells induced by ischemia and reperfusion).
  4. Figure 3. You should correct the spelling of gene and protein and add an explanation to the sign «grid». You should clarify that the number of squares corresponds to the number of replicates per group. Can the authors explain the reason for the different number of measurements at time points? So, for example, we take figure 3 (A). At point zero, the number of measurements is not clear. We have 5 measurements In the ischemia group (1 hour), in the ischemia group (3-12 hours) - 6 measurements. In the reperfusion group we have 4 measurements. Why? Add explanation for «****».
  5. Figure 3 (В). You need to specify the units of measure. Check in the text of the article whether the units of measurement are indicated everywhere on the graphs.
  6. Line 138. The authors are encouraged to designate transduced H9c2 cells with a different name from NEU3. This will avoid confusion because the NEU3 gene codes for Sialidase-3 in humans. This new designation should be entered in the text, in figure 4 and in the caption to it. This also applies to figures S2, 5, 6, S3, 7, 8, 9.
  7. Line 139. «….with a scrambled lentiviral vector and used as a control.» Add explanation for SCR .
  8. Figure 4. In the caption to the figure, when you describe a gene, you should use «Neu3» and when you describe a protein you should use «Sialidase-3». Add explanation for SCR and «****»
  9. Figure 5 (С). Indicate in the caption to the figure the designation of the image in the upper right corner in the SCR group.
  10. Figure 5. Indicate in the caption to the figure information about the meaning of the squares on the graphs. Does the number of squares in the group correspond to the number of replicates? Add explanation for «****»,CTRL and SCR.
  11. Figure 6. When you describe a gene, you should use «Neu3», and when you describe a protein - «Sialidase-3». You should edit it in the caption to the figure and in the figure. Indicate in the caption to the figure information about the meaning of the squares on the graphs. Does the number of squares in the group correspond to the number of replicates? Add explanation for «****».
  12. Line 205. You should decrypt the designation LR322 in the caption to the figure.
  13. Figure 7 (А). You should decrypt the designation «S» and «N». Indicate in the caption to the figure information about the meaning of the squares on the graphs. I believe that your addition of the Gapdh/Histone H3 immunoblot will improve your article (see comments to Paragraph 4.9). Add explanation for «****» and SCR.
  14. Figure 8. Comments as to figure 5. Add explanation for «CTRL».
  15. Figure S4. The authors should bring protein designations to a common form. You need to replace «Clx» with «Calnexin» (as in the main body of the article). I believe that your addition of the Gapdh/Histone H3 immunoblot will improve your article (see comments to Paragraph 4.9). It is necessary to explain what the lower panel means in letters (A) and (B).
  16. Figure 9. Comments as to figure 5. I believe that your addition of the Gapdh/Histone H3 immunoblot will improve your article (see comments to Paragraph 4.9).
  17. Line 372. Why did the authors choose 21% O2 air-humidified atmosphere? The authors did not take the numbers 25% or 50%, did they?
  18. Paragraph 4. I think that when you describe cell experiments you should add Cell Confluence Percentage Value. For example, cells that fill the culture dish by 50% will collect less protein and RNA than cells with 90-95% confluence.
  19. Paragraph 4.3. In addition, the authors should note whether they developed the primers themselves or was it ready-made. If the authors developed on their own, they should note what programs were used to develop and check the quality of primers (Primer-BLAST, PCR Primer Stats, Multiple Primer Analyzer)? After that authors can already refer to table 1, where primer sequences are indicated.
  20. Line 394. The gene in this line has not been fully described. The object of the study is a rat/mouse. Therefore, the genes were Rpl13a (RPL13A in humans) and Neu3 (NEU3 in humans). It is necessary to make these changes throughout the text (including figures) and in the Table 1. This applies to everything where the authors describe genes.
  21. The authors did not use the another housekeeper as an internal control, did they? Why? It could be Gapdh, B2m or Actb.
  22. Paragraph 4.4. and 4.9. It is necessary to add information about the kit with which the authors measured the amount of collected protein.
  23. Line 484. The Authors should decrypt the abbreviation HPTLC (High-performance thin-layer chromatography).
  24. Paragraph 4.5. I think that the authors had removed the cells from the cell dish before counting the cells on the Countess II…. What did the authors do with the cells after counting? The paragraph does not mention this.
  25. Paragraph 4.9. Is it possible for the authors to make a Western Blot using another loading control? The intensity of the Calnexin band in the samples is variable. The Authors can use, for example, Mitochondria Fraction Western Blot Cocktail (ab139416) as loading control (it contains Gapdh/histone H3).
  26. Paragraph 4.11. What time frame was used when you adjusted the concentration of LR322? If this is indicated, then figure 6 (A) becomes clear.
  27. Paragraph 4.12. I recommend to add a manufacturer for LY294002 and PD98059.
  28. Paragraph 4.14. It is necessary to indicate the total number of animals that were used in the experiment. There were 25 of them, right? In brackets, as an explanation, you can specify the number of animals per group. It is still not entirely clear to me how the animals were withdrawn from the experiment…. Five mice of the SHAM group were withdrawned after surgery after 30 minutes (in parallel with the ischemia group), right? The SHАM group was not used after 1, 4, and 7 days, right? It could control for these time points.
  29. Paragraph 4.15. The authors here described a technique for dividing the infarct area from the healthy cardiac tissue. Is it possible for the authors to insert an explanatory photograph in figure 1 (A)? In this case the origin of healthy tissue in Figure 1 (F) becomes clear.

The study is very interesting and can be published after revision of these points.

Reviewer 3 Report

Dear authors, the comments/suggestions were exhaustively addressed, I did appreciate the effort.

Author Response

We thank the reviewer for the suggestions that improved our manuscript.

Round 3

Reviewer 2 Report

I thank the authors for their responses to my comments. The research team has done a great job.

The authors should replace NEU3 with OX-Neu3 in Figure S2.